Effects of the Gully Land Consolidation Project on soil properties and carbon stocks of farmland in the Chinese Loess Plateau

Qiu Zeqiong 1
Gong Tingjun 2
Chen Xiaocao 2
Yu Xia yuxia@git.edu.cn 1 2 3
Wu Qixin qxwu@gzu.edu.cn 1
An Yanling 1 2
Zeng Jie 1
1 Key Laboratory of Karst Georesources and Environment (Guizhou University), Ministry of Education, College of Resources and Environmental Engineering, Guizhou University , Guiyang , China
2 College of Resources and Environmental Engineering, Guizhou Institute of Technology , Guiyang , China
3 State Key Laboratory of Environmental Geochemistry, Institute of Geochemistry, Chinese Academy of Sciences , Guiyang , China
Brygadyrenko Viktor
Electronic publication date: 2025 Jun 11
Publication date: 2025
Volume: 13
Electronic Location ID: e19540
Received 2025 Jan 31; Accepted 2025 May 8
Copyright: ©2025 Qiu et al.
Copyright year: 2025
Copyright holder: Qiu et al.
License: This is an open access article distributed under the terms of the Creative Commons Attribution License, which permits unrestricted use, distribution, reproduction and adaptation in any medium and for any purpose provided that it is properly attributed. For attribution, the original author(s), title, publication source (PeerJ) and either DOI or URL of the article must be cited.
License URL: https://creativecommons.org/licenses/by/4.0/

Keywords: Gully Land Consolidation, Soil properties, Soil carbon, Farmland management, Chinese Loess Plateau, Anthropic activities

Funding: Strategic Pilot Project (Class B) of the Chinese Academy of Sciences No. XDB4000000 Open Fund of State Key Laboratory of Loess and Quaternary Geology No. SKLLQG2036 Guizhou Provincial Science and Technology Support Program No. QianKeHeJiChu-ZK [2024] YiBan511 Research Initiation Program for High-level Talents of Guizhou Institute of Technology in 2023 No. 2023GCC082 This work was supported by the Strategic Pilot Project (Class B) of the Chinese Academy of Sciences (No. XDB4000000), the Open Fund of State Key Laboratory of Loess and Quaternary Geology (No. SKLLQG2036), the Guizhou Provincial Science and Technology Support Program (No. QianKeHeJiChu-ZK [2024] YiBan511), and the Research Initiation Program for High-level Talents of Guizhou Institute of Technology in 2023 (No. 2023GCC082). The funders had no role in study design, data collection and analysis, decision to publish, or preparation of the manuscript.

==============================
A megaproject named Gully Land Consolidation (GLC) was implemented in the Chinese Loess Plateau (CLP) to address food shortages. However, its effects on soil properties and carbon stocks remain unclear. This study evaluates the effects of the GLC project on soil properties and carbon stocks by analyzing the properties of 14 farmland soil profiles (0–500 cm depth) in the Gutun watershed (GT-T), a typical area treated by the GLC project. Soil particle size, water content (SWC), bulk density (BD), contents of soil organic carbon (SOC), inorganic carbon (SIC), and total carbon (TC) were measured, and carbon stocks were calculated. Data from the GT-T watershed were compared to untreated areas in the CLP (CLP-U) and outside the CLP (Other-U). The results showed that the silt proportion (76.0%), SWC (17.7 ± 5.2%), and BD (1.5 ± 0.2 g cm−3) in the 0–500 cm depth in the GT-T watershed was 1.1 to 2.1 times, 1.6 to 2.8 times, and 1.0 to 1.3 times higher than in untreated areas, respectively. However, the SOC stock (2.5 kg m−2) at 0–100 cm depth was 0.7 to 10.2 kg m−2 lower than in the untreated areas. These results indicate that although the GLC project can improve soil structure and water retention capacity, it may also lead to the loss of surface SOC, thereby reducing soil fertility. The protection and replenishment of surface SOC should be emphasized during the cultivation and management of the newly reclaimed farmland in GLC-treated areas. This study highlights the effects of the GLC project on soil properties and carbon stocks and provides valuable insights for optimizing land reclamation management in the CLP.

Introduction

Soil physical and chemical properties, such as bulk density (BD), porosity, particle size composition, and soil organic carbon (SOC) content, are important indicators for assessing soil quality. These properties reflect soil permeability, water retention capacity, nutrient supply ability, and fertility, which significantly influence the sustainability of land use (Wang & Shao, 2013; Zhang et al., 2020; Li et al., 2021). Understanding these properties forms the foundation for rational soil management and land use planning. Soil properties are determined by the soil formation process, which is highly complex and mainly influenced by five natural factors: parent material, climate, topography, biology, and time (Jenny, 1941). However, with the intensification of anthropic activities, anthropogenic factors have been gradually regarded as a sixth soil-forming factor, significantly impacting both soil formation processes and soil properties (Dazzi & Monteleone, 2007; Leguédois et al., 2016; Li et al., 2021). For example, land preparation and terrace construction significantly modify soil properties through multiple mechanisms: sloping terrace construction, topsoil stripping and backfilling, and mechanical compaction. These activities alter soil horizon thickness, soil structure, weathering rates, and organic matter mineralization processes, ultimately affecting the redistribution of soil moisture and nutrients (Wang et al., 2016; Tang et al., 2019; Ma et al., 2020; Li et al., 2021). Therefore, the effects of anthropic activities on soil properties should not be overlooked.

The Chinese Loess Plateau (CLP) is one of the most severely eroded areas in the world, ecologically vulnerable and sensitive to human activity (Fu et al., 2017). The Chinese government launched the “Grain for Green Program” (GGP) in the CLP in 1999 to control soil erosion (Fang et al., 2012). However, the large-scale vegetation rehabilitation under the program has resulted in a substantial decrease in hillslope farmland, leading to farmland shortages and an imbalance between food supply and demand (Shi et al., 2020). To offset the loss of farmland and address food shortages, a megaproject, the “Gully Land Consolidation” (GLC), was launched in Shaanxi Province in 2011. The main approach of the GLC project includes reshaping creek valleys by incising foot slopes, filling stream channels and ditches, constructing drainage canals, dams and reservoirs, and creating flat farmland (Jin et al., 2019; Ma et al., 2020). From 2013 to 2017, the project created 3.33 × 104 ha of farmland in Yan’an City (Jin et al., 2019), significantly improving regional agricultural conditions. In recent years, numerous studies have investigated the effects of the GGP on soil properties (Li et al., 2018; Yang et al., 2020; Song et al., 2021). These studies show that the GGP increased SOC storage (An et al., 2010; Song et al., 2021), enhanced soil aggregate stability (An et al., 2010), and reduced soil BD (Zhang et al., 2018). However, it also resulted in significant soil moisture loss, exacerbating soil drought (Yang et al., 2020). In contrast, limited research has been conducted on the effects of the GLC project on soil properties and carbon stocks of farmland, leaving a gap in effective guidance for managing newly reclaimed farmland.

In this study, we measured the farmland soil particle size, water content (SWC), bulk density (BD), organic carbon (SOC), inorganic carbon (SIC), total carbon (TC) content, and estimated the stocks of SOC (SOCS), SIC (SICS), and TC (TCS) in the Gutun watershed (GT-T), a typical area treated by the GLC project. These indicators were compared with those from untreated areas in the CLP (CLP-U) and outside the CLP (Other-U). The objectives of this study were: (1) investigate the current status of soil properties and carbon stocks within the 0–500 cm soil profile of newly constructed farmland following the GLC project treated, (2) clarify the differences in soil properties and carbon stocks of farmland between the GLC-treated and untreated areas, and (3) evaluate the effects of the GLC project on soil properties and carbon storage.

Materials & Methods

Description of the study area

The Gutun watershed (36°46′39″–37°3′34″N, 109°41′02″–109°56′58″E) is located 46 km east of Yan’an City, Shanxi Province (Fig. 1). It covers approximately 24 km2 and has an average elevation ranging from 956 to 1220m above sea level. The region is characterized by a continental monsoon climate with an annual mean temperature and precipitation of 9.7 °C and 541 mm, respectively. 70% of the precipitation falls between June and September (Wang et al., 2008; Yu et al., 2019a; Zhao et al., 2019b). According to the World Reference Base for Soil Resources (WRB) classification system, the soils in the watershed primarily belong to “Calcic Cambisols”, which are typical soils of the CLP.

Figure 1 Soil sampling sites of farmland in the Gutun Watershed.

These farmland include five cultivation ages: R5 (cultivated for 5 years, closed green/black circle), R15 (cultivated for 15 years, yellow triangle), R35 (cultivated for 35 years, green star), R60 (cultivated for 60 years, closed green circle with centered black point), R70 (cultivated for 70 years, closed green circle).

The Gutun watershed is not only a typical representation of the gully terrain in the CLP but also a successful representation of the GLC project. According to field surveys, as early as the 1940s, local farmers undertook small-scale land reclamation through constructing sluice ditches and incising slopes to fill gullies, in order to expand arable land and enhance grain production. In 2011, the government officially named these land development measures “Gully Land Consolidation” (GLC) and initiated large-scale implementation. As the GLC project progressed over different periods, a series of newly reclaimed farmland with varying cultivation ages has been established in the watershed, including farmland cultivated for 5 years (R5), 15 years (R15), 35 years (R35), 60 years (R60), and at least 70 years (R70). The cultivation ages of these farmland were verified through interviews with local elders. Except for the R35, which was planted as a forest nursery, the others were planted with maize (Yu et al., 2019a).

Sample collection and laboratory analysis

To ensure that the collected soil samples represented all farmland treated by the GLC project in the Gutun watershed, we collected soil samples from farmland with varying cultivation ages, all of which have been artificially created since the 1940s. Soil samples were collected from three 1 m × 1 m quadrats in each type of reclaimed farmland located by GPS from March to April 2017, except R70, because only two 70-year-old farmland were available. Before sampling, all litter was removed from each sampling site. Every site was excavated to a depth of 500 cm until the calcification or overflow layer was reached (Yu et al., 2019a). Undisturbed soil samples were collected using 275 soil core rings with a diameter of five cm. BD was measured every 10 cm from 0 to 20 cm depth and every 20 cm from 20 to 500 cm depth. Another 275 samples were also collected to measure SWC, particle size, TC, and SIC content.

BD was calculated based on the volume-mass relationship of each soil sample dried to a constant weight at 105 °C (Wang, Shao & Liu, 2013). The calculation formula is as follows: (1) BD=MdVd

where, Md and Vd represent the dry mass and volume of the soil after being dried to a constant weight, respectively.

A small portion of each soil sample was dried at 105 °C to a constant weight, and SWC was determined based on the mass loss before and after drying (Zhao et al., 2019b). The remaining soil samples were air-dried for the particle size, TC, and SIC content analysis.

Weighed air-dried soil of approximately 0.2 g and added 10 mL of 10% H2O2, then heated on a 95 °C hot plate to remove organic matter until bubbling ceased. Next, 10% hydrochloric acid (HCl) was added to remove carbonates, continuing until no further bubbling occurred. The sample was then washed with distilled water until neutral, left to stand for 12 h, and the supernatant was decanted. Finally, 10 mL of 1 mol L−1 sodium hexametaphosphate was added, and the sample was sonicated for 10 min. Soil particle size was measured using a Mastersizer 3000 (Malvern Instruments, England). Soil particles were classified as clay (<2 µm), silt (2–50 µm), and sand (50–2,000 µm) based on particle size.

The remaining air-dried soil samples were sieved through a 2 mm mesh. 10 mg of each sieved sample was weighed, and wrapped in aluminum foil. TC content was measured using an elemental analyzer at a combustion temperature of 950 °C (Vario EL cube, Elemental Analyzer, Elementar, Germany). SIC refers to the carbonate minerals in the soil, such as CaCO3 and MgCO3 (Wu et al., 2009; Liu et al., 2017), it can be determined by measuring the CO2 released during acid hydrolysis of the soil sample. Approximately 10 g of sieved soil was placed in a Y-tube, with 2 mL of concentrated phosphoric acid added to the other side. The tube was sealed and vacuumed to 10−3 atm, allowing the sample and phosphoric acid to mix. The mixture was heated at 110 °C for 1 h on a hot plate, and the CO2 released during the process was measured using a vacuum line. The SIC content was calculated from the CO2 released. The SOC content was obtained by subtracting SIC from TC (Yu et al., 2019a). All experiments were conducted at the Xi’an Accelerator Mass Spectrometry Center.

Calculation of carbon stocks

Carbon stock, i.e., carbon mass per unit area for a given depth, was calculated according to the following equations (Grimm et al., 2008): (2) SOCstock=SOC×BD×1−Fcontent×Δd×0.01

(3) SICstock=SIC×BD×1−Fcontent×Δd×0.01

(4) TCstock=SOCstock+SICstock

where, SOCstock, SICstock, and TCstock represent SOC, SIC, and TC stock (kg m−2), respectively; SOC and SIC represent SOC and SIC content (g kg−1), respectively; BD is soil bulk density (g cm−3); Fcontent is the proportion of non-soil particles > 2 mm diameter (dimensionless). The occurrence of coarse fragments in the loessal soil of the CLP is rare, so Fcontent is usually negligible; Δd is the thickness of the layer (cm). Carbon stock per unit area for a given depth was calculated by summing carbon stock over all layers.

Data classification and sources

The soil data of farmlands analyzed in this study were categorized into three groups based on whether the study area had been treated by the GLC project: (1) data from areas treated by the GLC project in the Gutun watershed (GT-T), (2) data from untreated areas within the CLP (CLP-U), and (3) data from untreated areas outside the CLP (Other-U). Detailed information is provided in Table 1.

Table 1 The soil properties and carbon stocks of farmland in the GT-T watershed, and the comparison with the CLP-U and Other-U areas.

Study area	Area type	Depth (cm)	Sand (%)	Silt (%)	Clay (%)	SWC (%)	BD (g cm −3 )	SOCS (kg m −2 )	SICS (kg m −2 )	STCS (kg m −2 )	SICS: SOCS ration	Reference	
GT-T watershed	GT-T watershed	0–100	24.0 ± 5.1	73.5 ± 4.9	2.5 ± 0.5	14.3 ± 3.7	1.4 ± 0.2	2.5	19.0	21.5	7.6	This study	
GT-U watershed	CLP-U area	0–100	—	—	—	10.8 ± 1.6	1.4 ± 0.1	—	—	—	—	Zhao et al. (2019b)	
Linfen-Yanchi climatic zones	CLP-U area	0–100	—	—	—	—	—	3.2	18.5	21.7	5.7	Han et al. (2018)	
Huining	CLP-U area	0–100	—	—	—	—	1.2	—	—	—	—	Zhang et al. (2018)	
Lanzhou	CLP-U area	0–100	—	—	—	—	—	6.1	20.8	26.9	3.4	Zhang et al. (2015)	
Yangling	CLP-U area	0–100	—	—	—	—	—	9.3	9	18.3	1.0	Wang et al. (2014)	
Jiegou watershed	CLP-U area	0–100	—	—	—	11.0 ± 1.2	—	—	—	—	—	Yu et al. (2019b)	
Longtan watershed	CLP-U area	0–100	—	—	—	8.3 ± 1.5	—	—	—	—	—	Yang et al. (2012)	
Laoyemanqu watershed	CLP-U area	0–100	35.0 ± 8.7	53.5 ± 7.1	11.6 ± 1.6	9.3 ± 0.5	—	—	—	—	—	Wang, Zeng & Zhong (2016)	
Yanqi Basin	Other-U area	0–100	20.1 ± 5.8	72.3 ± 4.4	7.6 ± 1.6	—	1.3 ± 0.1	9.2	42.0	51.2	4.6	Wang et al. (2015)	
Urumqi	Other-U area	0–100	—	—	—	—	—	9.2	5.4	14.6	0.6	Wang et al. (2014)	
Hebei Plain	Other-U area	0–100	17.0	66.0	17.0	—	—	7.5	16.5	24.0	2.2	Shi et al. (2017)	
The upper Yellow River Delta	Other-U area	0–100	—	—	—	—	—	5.7	16.9	22.6	3.0	Guo et al. (2016)	
Quzhou	Other-U area	0–100	—	—	—	—	—	6.7	11.3	18	1.7	Bughio et al. (2016)	
Zhengzhou	Other-U area	0–100	—	—	—	—	—	6.6	8.1	14.7	1.2	Wang et al. (2014)	
Northeast China Plain	Other-U area	0–100	—	—	—	15.5 ± 5.7	1.4 ± 0.1	12.7	5.4	18.1	0.4	Zhu et al. (2023)	
Lishu	Other-U area	0–100	—	—	—	—	—	4.9	3.6	8.5	0.7	Xie et al. (2023)	
Sanjiang Plain	Other-U area	0–100	—	—	—	—	1.3	—	—	—	—	Man et al. (2017)	
GT-T watershed	GT-T watershed	0–300	22.7 ± 2.2	74.5 ± 1.9	2.8 ± 0.3	16.5 ± 2.2	1.5 ± 0.1	8.3	55.0	64.2	6.6	This study	
GT-U watershed	CLP-U area	0–300	—	—	—	10.6 ± 0.9	1.4 ± 0.0	—	—	—	—	Zhao et al. (2019b)	
Linfen-Yanchi climatic zones	CLP-U area	0–300	62.7 ± 10.5	31.9 ± 10.5	4 ± 0.9	—	1.2 ± 0.0	7.5	57.1	64.6	7.6	Han et al. (2018)	
Jiegou watershed	CLP-U area	0–300	—	—	—	11.0 ± 0.4	—	—	—	—	—	Yu et al. (2019b)	
Longtan watershed	CLP-U area	0–300	—	—	—	8.7 ± 0.5	—	—	—	—	—	Yang et al. (2012)	
Laoyemanqu watershed	CLP-U area	0–300	33.5 ± 7.2	54.6 ± 5.9	11.9 ± 1.4	9.0 ± 0.6	—	—	—	—	—	Wang, Zeng & Zhong (2016)	
GT-T watershed	GT-T watershed	0–500	22.1 ± 5.5	75.1 ± 5.0	2.8 ± 0.7	17.7 ± 5.2	1.5 ± 0.2	14.6	83.8	98.4	5.7	This study	
GT-U watershed	CLP-U area	0–500	25.6	69.7	4.7	10.5 ± 0.7	—	—	—	—	—	Zhao et al. (2019a), Zhao et al. (2019b)	
Jiegou watershed	CLP-U area	0–500	—	—	—	11.4 ± 0.7	—	—	—	—	—	Yu et al. (2019b)	
Longtan watershed	CLP-U area	0–500	—	—	—	9.8 ± 1.3	—	—	—	—	—	Yang et al. (2012)	
Laoyemanqu watershed	CLP-U area	0–500	34.1 ± 6.4	54.0 ± 5.3	11.9 ± 1.2	8.7 ± 0.8	—	8.3 ± 2.3	—	—	—	Wang, Zeng & Zhong (2016)	
Fufeng	CLP-U area	0–500	40.5	35.8	23.6	—	—	23.1	115.0	138.1	5.0	Song et al. (2020)	
Yongshou	CLP-U area	0–500	35.2	38.9	26.3	—	—	18.1	2.0	20.1	0.1	Song et al. (2020)	
Notes.

—represents no data is available.

The CLP-U areas include the untreated areas in the Gutun watershed (GT-U), Linfen-Yanchi climatic zones, Huining, Lanzhou, Yangling, Jiegou watershed, Longtan watershed, Laoyemanqu watershed, Fufeng, and Yongshou. The farmland soil data from these areas were obtained from relevant references in Table 1.

The Other-U areas include the Yanqi Basin, Urumqi, Hebei Plain, the upper Yellow River Delta, Quzhou, Zhengzhou, Northeast China Plain, Lishu, and Sanjiang Plain. The farmlands soil data from these areas were also obtained from relevant references in Table 1.

Statistical analysis

Basic data processing was conducted using Microsoft Excel 2021. The sampling map was drawn using ArcMap 10.2. Statistical significance of differences among certain parameters was tested using one-way analysis of variance (ANOVA) in SPSS 26.0, with a significance level set at p < 0.05. Data visualization was performed using Origin 2025.

Results and Discussion

The GLC project changed the farmland soil texture

The mean proportions of sand, silt and clay in the farmland soil at depths of 0–500 cm in the GT-T watershed were 21.0%, 76.0% and 3.0% respectively (Fig. 2A), and the soil texture was predominantly silty loam (Fig. 2B). From R5 to R60, the sand proportion increased from 21.1% to 30.5%, while the silt and clay proportions decreased from 76.0% and 3.0% to 67.3% and 2.2%, respectively. By R70, the proportions of clay, silt, and sand had returned to the levels observed in R5 (p > 0.05) (Table 2). The proportions of sand, silt, and clay in the 0–100 cm layer were 24.5%, 73.0%, and 2.5%, respectively. In the 0–300 cm layer, these proportions were 22.7%, 74.5%, and 2.8%, and in the 0–500 cm layer, they were 21.5%, 75.6%, and 2.9%, respectively. With increasing depth, the proportion of sand decreased, while the proportions of silt and clay increased (Table 3).

Figure 2 Comparative analysis of soil texture and particle size distribution at different depths.

The soil particle size distribution (A) and soil texture (B) of farmland in the GT-T watershed in this study, along with the comparative data from farmland in the CLP-U and Other-U areas. The references for the data from farmland in the CLP-U and Other-U areas are the same as in Table 1.

Table 2 Soil properties of farmland at 0–500 cm depths in the GT-T watershed.

Cultivation ages	Texture	BD (g cm−3)	SWC (%)	TC (g kg−1)	TCS (kg m−2)	
	Sand (%)	Silt (%)	Clay (%)					
R5	21.1 ± 4.0 bc	76.0 ± 3.8 ab	3.0 ± 0.5 ab	1.6 ± 0.1 a	17.3 ± 3.4 a	14.9 ± 2.0 ab	117.7	
R15	20.2 ± 4.6 c	76.9 ± 4.3 a	2.8 ± 0.6 bc	1.5 ± 0.1 a	19.8 ± 3.9 a	12.0 ± 5.3 c	76.2	
R35	23.9 ± 4.5 b	73.6 ± 4.2 b	2.5 ± 0.4 cd	1.2 ± 0.1 c	13.9 ± 2.5 b	15.9 ± 1.4 a	97.3	
R60	30.5 ± 3.5 a	67.3 ± 3.3 c	2.2 ± 0.5 d	1.4 ± 0.1 b	19.2 ± 7.9 a	15.1 ± 1.7 ab	106.8	
R70	19.9 ± 6.5 c	76.8 ± 5.9 a	3.2 ± 0.9 a	1.6 ± 0.2 a	18.1 ± 2.7 a	14.3 ± 3.1 b	94.2	
Notes.

Lowercase letters represent differences in different layers (p < 0.05).

Table 3 The variation of farmland soil properties and carbon stocks with depth in the GT-T watershed.

Depth (cm)	Sand (%)	Silt (%)	Clay (%)	SWC (%)	BD (g cm −3 )	SOC ( g kg −1 )	SIC ( g kg −1 )	TC ( g kg −1 )	SOCS (kg m −2 )	SICS (kg m −2 )	TCS (kg m −2 )	SICS: SOCS ration	
0–100	24.0 ± 5.1 a	73.5 ± 4.9 c	2.5 ± 0.5 b	14.3 ± 3.7d	1.4 ± 0.2 a	1.9 ± 1.2 a	13.3 ± 2.0 a	15.2 ± 2.1 a	2.5 ± 0.7 a	19.0 ± 2.9 a	21.5 ± 2.7 a	7..6	
100–200	22.6 ± 5.6 ab	74.6 ± 5.1 bc	2.8 ± 0.7 ab	16.9 ± 4.5 c	1.5 ± 0.2 a	2.0 ± 1.1 a	12.7 ± 1.8 ab	14.6 ± 1.7 a	3.0 ± 1.1 a	18.6 ± 2.4 a	21.6 ± 2.9 a	6.2	
200–300	20.7 ± 6 bc	76.3 ± 5.5 ab	3.0 ± 0.8 a	19.1 ± 4.3 b	1.5 ± 0.2 a	1.9 ± 0.9 a	12.0 ± 3.5 ab	13.9 ± 3.5 a	2.8 ± 1.1 a	17.4 ± 2.7 a	20.2 ± 2.8 a	6.2	
300–400	20.3 ± 5.1 bc	76.8 ± 4.7 ab	3.0 ± 0.5 a	20.1 ± 4.9 b	1.5 ± 0.2 a	2.3 ± 1.1 a	11.5 ± 4.0 b	13.9 ± 4.5 a	3.4 ± 1.6 a	16.5 ± 5.9 a	19.9 ± 6.7 a	4.9	
400–500	19.9 ± 3.5 c	77.1 ± 3.1 a	3 ± 0.6 a	22.5 ± 5.0 a	1.4 ± 0.2 a	2.2 ± 1.3 a	11.3 ± 5.0 b	13.6 ± 5.6 a	2.9 ± 1.1 a	14.8 ± 7.2 a	17.8 ± 7.8 a	5.1	
Mean	22.1 ± 5.5	75.1 ± 5.0	2.8 ± 0.7	17.7 ± 5.2	1.5 ± 0.2	2.0 ± 1.1	12.4 ± 3.2	14.4 ± 3.4	14.6 ± 3.7	83.8 ± 13.1	98.4 ± 15.4	5.7	
Notes.

Lowercase letters represent differences in different layers (p < 0.05).

As shown in Fig. 2A, the proportion of silt in the 0–100 cm (74.0%), 0–300 cm (74.0%), and 0–500 cm (76.0%) soil layers was the highest in the GT-T watershed compared to untreated areas (the CLP-U and Other-U), which was 1.0 to 1.4, 1.3 to 2.2, and 1.1 to 2.1 times higher than in these areas, respectively. In the 0–100 cm layer, the soil texture of farmland in the GT-T watershed was similar to that in the Hebei Plain, Yanqi Basin, and Laoyemanqu watershed, all of which were silty loam (Fig. 2B). In the 0–300 cm profile, the soil texture of farmland in Linfen-Yanchi climatic zones of the CLP-U areas was sandy loam (Fig. 2B), which differed significantly from that in the GT-T watershed. In the 0–500 cm profile, the soil texture of farmland in the GT-T, GT-U, and Laoyemanqu watersheds were silt loam, with the GT-T watershed exhibiting the highest silt proportion and the lowest proportions of clay and sand (Fig. 2B). Compared to farmland in the Guanzhong areas of the CLP, such as Yongshou and Fufeng (loam), significant differences in soil texture were observed (Fig. 2B).

The results above demonstrate that the GLC project has significantly altered soil composition in newly reclaimed farmland, manifesting as decreased proportions of clay and sand alongside increased silt content. This textural evolution has led to a trend in soil texture shifting from silt loam to silt, thereby enhancing soil and water conservation capacity. These textural modifications were primarily attributed to anthropogenic disturbances during loess backfilling operations and subsequent mechanical compaction processes. Specifically, these anthropogenic engineering practices disintegrated soil coarse aggregates into finer particulates, thereby inducing a decrease in both soil porosity and particle diameter (Ma et al., 2020).

The GLC project enhanced the water conservation capacity of deeper soil layer

There were no significant differences in SWC at depths of 0–500 cm among R5 (17.3 ± 3.4%), R15 (19.8 ± 3.9%), R60 (19.2 ± 7.9%), and R70 (18.1 ± 2.8%) (p > 0.05), but they were significantly higher than R35 (13.9 ± 25 g kg−1) (p < 0.05) (Table 2). The SWC in the 0–100 cm layer of farmland in the GT-T watershed was 14.3 ± 3.7%, slightly lower than that in the Northeast China Plain (15.5 ± 5.7%), but higher than that in the CLP-U areas, including the GT-U, Jiegou, Longtan, and Laoyemanqu watersheds (Table 1, Fig. 3A). The SWC in the 0–300 cm and 0–500 cm depths of farmland in the GT-T watershed were 16.5 ± 2.2% and 17.7 ± 5.2%, respectively, which was 1.1–1.9 times and 1.1–2.0 times higher than that in the corresponding depths of farmland in the CLP-U areas (Fig. 3A). Thus, across the entire 500 cm depth, the SWC in the GT-T watershed remained at a higher level.

Figure 3 Comparative analysis of SWC at different depths.

SWC values at different depths (A) and their variations in the 0–500 cm profile (B) of farmland in the GT-T watershed in this study, along with the comparative data from farmland in the CLP-U and Other-U areas. The references for the data from farmland in the CLP-U and Other-U areas are the same as in Table 1.

To investigate the vertical profile variation of SWC in the 0–500 cm profile of farmland in the GT-T watershed, the SWC at each 100 cm depth interval was analyzed and compared with that in the CLP-U areas, as shown in Fig. 3B. SWC in the GT-T watershed increased with depth and was 1.2 to 1.6 times higher in the 100–500 cm layers compared to the surface (0–100 cm), showing significantly greater moisture retention than CLP-U areas (Fig. 3B, Table 3). In contrast, in the CLP-U areas, such as the Longtan and Laoyemanqu watersheds, SWC decreased with increasing depth, while the GT-U and Jiegou watersheds showed little variation. The mean SWC across the 0–500 cm profile in the GT-T watershed (18.6%) was 1.6–2.8 times higher than that in the CLP-U areas, such as the GT-U (10.5%), Jiegou (11.3%), Longtan (6.6%), and Laoyemanqu (8.3%) watersheds.

In the GT-T watershed, soil water storage in the 100–500 cm layer (2,606 mm) accounted for 85% of the total soil water storage (3,054 mm) in the 0–500 cm depth. This indicated that the GLC project promoted greater water stored in the deeper soil (100–500 cm). This may be attributed to the land in the gully, filled to a depth of approximately 2 m by the GLC project, which not only reduced the residence time of runoff from the adjacent slopes, but also decreased the runoff volume and velocity within the gully, thereby promoting more water to infiltrate and be stored in the deeper soil. Additionally, a significant negative correlation was observed between SWC and the sand proportion, while a significant positive correlation was found with the silt proportion (Fig. 4). These results suggest that coarser soil particles in the surface layers hinder water conservation. In contrast, a higher silt proportion in the deeper layers provided favorable conditions for soil water storage after the GLC project implementation.

Figure 4 Spearman rank correlation analysis of soil properties and carbon of farmland in the GT-T watershed.

The color intensity represents the positive correlation (red) or negative correlation (blue), and the darker color represents the greater absolute value of correlation coefficient r, * represents p < 0.05, ** represents p < 0.01.

The GLC project increased soil BD in farmland

Soil BD in the 0–500 cm depth of farmland in the GT-T watershed varied from 1.0 to 1.8 g cm−3 (mean = 1.5 ± 0.2 g cm−3). Significant differences in BD were observed among farmland with different cultivation ages (p < 0.05). The lowest BD value was found in R35 (1.2 ± 0.1 g cm−3), while R70 exhibited the highest BD value (1.6 ± 0.2 g cm−3) (Table 2). However, no significant variations in BD were observed with increasing depth (p > 0.05) (Table 3).

The BD of 0–100 cm (1.4 g cm−3) and 0–300 cm (1.5 g cm−3) soil layers in the GT-T watershed was 1.0 to 1.2 and 1.1 to 1.3 times higher than in untreated areas, respectively. Specifically, the BD of the 0–100 cm soil layer in the GT-T watershed was similar to that in the GT-U watershed (1.4 g cm−3) and Northeast China Plain (1.4 g cm−3), but higher than that in the Other-U areas, such as Yanqi Basin (1.3 g cm−3), Sanjiang Plain (1.3 g cm−3), and Huining county (1.2 g cm−3) (Fig. 5A, Table 1). In the 0–300 cm, the mean BD was slightly higher in the GT-T watershed than that in the GT-U watershed (1.4 g cm−3), and significantly higher than that in the Linfen-Yanchi climatic zones (1.2 g cm−3) (Fig. 5A, Table 1). The variation in BD with depth in the GT-T and GT-U watersheds was analyzed (Fig. 5B). It was observed that the mean BD in the 0–100 cm layer in the GT-T watershed was slightly higher than that in the GT-U watershed. However, this difference became significantly more pronounced in the 100–300 cm layer.

Figure 5 Comparative analysis of BD at different depths.

BD values at different depths (A) and their variations in 0–500 cm profile (B) of farmland in the GT-T watershed in this study, along with the comparative data from farmland in the CLP-U and Other-U areas. The references for the data from farmland in the CLP-U and Other-U areas are the same as in Table 1.

The higher soil BD values in the GT-T watershed can be attributed to soil compaction during the GLC project, caused by machinery (Håkansson & Lipiec, 2000; Hamza & Anderson, 2005; Lu et al., 2019). During compaction, larger soil particles were broken into smaller particles, reducing the pore space between particles and resulting in denser soil. As shown in Fig. 4A, a significant negative correlation between BD value and sand proportion, while a significant positive correlation with silt and clay proportions, further supports the idea that soils with smaller particles are more compact.

Is a higher BD value an advantage or a disadvantage? According to the study by Dong et al. (2021), a BD value of 1.5 g cm−3 was considered optimal for cultivated land, as it promoted crop growth and optimal yield. In this study, the mean BD values of the 0–100 cm layer (1.4 g cm−3) and 0–300 cm layer (1.5 g cm−3) in the GT-T watershed were closer to this optimal value compared to farmland in the CLP-U and Other-U areas, which may be more favorable for crop growth and achieving optimal yield. However, the relationship between BD and soil health remains debated, because some studies indicated that lower BD values typically reflected better soil structure, facilitating gas exchange and improving permeability (Zhang et al., 2018; Lu et al., 2019). Therefore, further investigation is necessary to determine the optimal BD value for farmland in the GT-T watershed.

The GLC project increased the SICS: SOCS ratio in the 0–100 cm soil layer

In the 0–500 cm soil profile, the TC contents of farmland in the GT-T watershed, ranked from high to low were as follows: R35 (15.9 g kg−1) > R5 (14.9 g kg−1) > R60 (15.1 g kg−1) > R70 (14.3 g kg−1) > R15 (12.0 g kg−1) (Table 2). The mean TC content was 14.4 ± 3.4 g kg−1, with SOC and SIC contents of 2.0 ± 1.1 g kg−1 and 12.4 ± 3.2 g kg−1, respectively. The TCS values of 0–500 cm depth ranked from high to low were as follows: R5 (117.7 kg m−2) >R60 (106.8 kg m−2) > R35 (97.3 kg m−2) > R70 (94.2 kg m−2) > R15 (76.2 kg m−2) (Table 2). The mean TCS value was 98 ± 15.4 kg m−2, with SOCS and SICS values of 14.6 ± 3.7 kg m−2 and 83.8 ± 13.1 kg m−2, respectively. The SICS:SOCS ratio was 5.7, indicating that the carbon storage was predominantly in the form of SIC (Table 3). With increasing depth, both SICS and TCS decreased at each 100 cm layer, while the SOCS of each soil layer below 100 cm was higher than that in the 0–100 cm layer (Table 3).

In the 0–100 cm layer, SOCS and SICS in the GT-T watershed were 2.5 kg m−2 and 19.0 kg m−2, respectively. Compared to the CLP-U and Other-U areas, SOCS of farmland in the GT-T watershed was the lowest, while SICS was relatively higher (Fig. 6A). In the 0–300 cm layer, SOCS and SICS in the GT-T watershed were 8.7 kg m−2 and 55.5 kg m−2, respectively, similar to the values found in Linfen-Yanchi climatic zones (Fig. 6A). In the 0–500 cm layer, SOCS and SICS in the GT-T watershed were 14.6 kg m−2 and 83.8 kg m−2, respectively, showing significant differences with that in Guanzhong areas, such as Yongshou and Fufeng (Fig. 6A). The TCS in the 0–300 cm and 0–500 cm layers in the GT-T watershed were 3 times and 4.6 times higher than that in the 0–100 cm layer, respectively. The SOCS in the GT-T watershed was generally lower than that in the CLP-U and Other-U areas, except for the Linfen-Yanchi climatic zones in the 0–300 cm soil layer.

Figure 6 Comparative analysis of soil carbon stocks and SICS: SOCS ratios at different depths.

Soil carbon stocks (A) and SICS: SOCS ratio (B) of farmland in the Gutun watershed in this study, along with comparative data from farmland in the CLP-U and Other-U areas. The references for the data from farmland in the CLP-U and Other-U areas are the same as in Table 1.

In the 0–100 cm layer, the SICS:SOCS ratio reached 7.6 in the GT-T watershed, which was significantly higher than in deeper layers and all comparison areas (Fig. 6B). This can be attributed to the lower SOCS values. Compared to the CLP-U areas (such as Yangling, Lanzhou, and Linfen-Yanchi climatic zones) and Other-U areas (such as Lishu, the Northeast China Plain, Zhengzhou, Quzhou, the upper Yellow River Delta, the Hebei Plain, Urumqi, and the Yanqi Basin), the SOCS in the 0–100 cm depth in the GT-T watershed decreased by 0.7–10.2 kg m−2 (Fig. 6A), but the SICS:SOCS ratio was 1.3-19 times higher than that in these areas (Fig. 6B). These results indicated that SOC loss occurred in the 0–100 cm soil layer of farmland in the GT-T watershed, leading to a significant increase in the SICS:SOCS ratio. Furthermore, compared to the SOC content in the 0–100 cm layer in the GT-U watershed (3.2 ± 0.4 g kg−1) (Zhao, 2022), the SOC content in the GT-T watershed (2.0 ± 1.1 g kg−1) decreased by 1.3 g kg−1, further confirming that the implementation of the GLC project resulted in SOC loss in the 0–100 cm layer.

The SOC loss in the 0–100 cm layer in the GT-U watershed was likely due to the heavy machinery compaction of backfilled soil during the implementation of the GLC project, which disrupted soil macroaggregates in the surface layer. The disruption of macroaggregates increased the contact between soil organic matter and microorganisms, thereby accelerating the mineralization of SOC. Additionally, the insufficient input of external organic matter during the geomorphic reshaping process further contributed to the significant loss of SOC in the surface soil (Ahirwal & Maiti, 2016; Liu et al., 2024). The SOC loss may indicate a reduction in soil organic matter and available nutrients (Liu et al., 2024), which is detrimental to crop growth. Therefore, in the management of newly reclaimed farmland, special attention should be given to the protection and replenishment of SOC in the 0–100 cm soil layer.

Conclusions

This study reveals that the GLC project significantly altered soil physical properties and carbon dynamics in the CLP. The project decreased the sand proportion and increased the silt proportion, making the soil texture more similar to silt soil. Furthermore, it enhanced the subsoil water retention capacity and brought the soil BD closer to the optimal value for crop growth. However, it also resulted in substantial loss of SOC in the surface layer (0–100 cm). These findings suggest that while the GLC project has advantages in improving soil structure and water retention, it may compromise long-term soil fertility if SOC loss is not addressed. Future reclamation projects should focus on protecting and replenishing surface SOC to maintain soil fertility and agricultural sustainability. This study provides valuable scientific guidance for land management and agricultural production in the CLP. It should be noted that this study was conducted in a single watershed affected by the GLC project, with limited spatial coverage. Therefore, the findings cannot be directly extrapolated to broader regional or national contexts. Future research should consider conducting similar studies in various communities or regions to contrast and validate the findings, and to further assess the generalizability and long-term impacts of the GLC project.

Supplemental Information

Supplemental Information 1 Raw data for Figs. 2–6 and Tables 2-3

We sincerely thank the editors and reviewers for their valuable comments on improving the manuscript.

Additional Information and Declarations

Competing Interests

Author Contributions

Data Availability

The authors declare there are no competing interests.

Zeqiong Qiu conceived and designed the experiments, performed the experiments, analyzed the data, prepared figures and/or tables, authored or reviewed drafts of the article, and approved the final draft.

Tingjun Gong performed the experiments, analyzed the data, prepared figures and/or tables, and approved the final draft.

Xiaocao Chen performed the experiments, analyzed the data, prepared figures and/or tables, and approved the final draft.

Xia Yu conceived and designed the experiments, authored or reviewed drafts of the article, and approved the final draft.

Qixin Wu conceived and designed the experiments, authored or reviewed drafts of the article, and approved the final draft.

Yanling An conceived and designed the experiments, authored or reviewed drafts of the article, and approved the final draft.

Jie Zeng conceived and designed the experiments, authored or reviewed drafts of the article, and approved the final draft.

The following information was supplied regarding data availability:

The raw measurements are available in the Supplementary File.

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
