# Peer review of "Effects of the Gully Land Consolidation Project on soil properties and carbon stocks of farmland in the Chinese Loess Plateau"

_PeerJ, doi:10.7717/peerj.19540_

## Round 0.1 · original submission · Major Revisions

Dear Dr. Yu, unfortunately, it currently contains many shortcomings and needs to be improved according to the reviewers' comments. I hope that you can improve each section of the article so that the article can be published.

Reviewer 1 ·

Basic reporting

.

Experimental design

.

Validity of the findings

No comments

Additional comments

This article makes a comparative analysis through a large number of data to assess the effects of the Gully Land Consolidation Project on soil properties and carbon stocks of farmland in the Gutun watershed, a typical gully area and a successful representation treated by the GLC project in the Chinese Loess Plateau. The article exhibits clarity in its research objectives, rich content, in-depth discussion, and concise yet provocative conclusions. It centers around soil properties and carbon stocks of newly cultivated farmlands after the implementation of the GLC project in the Chinese Loess Plateau, a topic of current relevance and interest. It underscores the profound influence of human activities on soil properties and carbon stocks and provides valuable insights for optimizing land reclamation management in the Chinese Loess Plateau. I think this manuscript can be accepted for publication after appropriate revisions according to the following suggestions:
1.The article contains some language errors, highlighting the necessity for additional proofreading to facilitate easy comprehension of information by readers.
2.Lines 55 - 60: The conjunction “and” is used several times in these sentences. It is recommended to further condense the text, refine the language, and improve readability.
3.Lines 68: “Shaanxi” is misspelled. Please check if “3.33 × 104 ha” should be “3.33 × 104 ha”.
4.Lines 90 - 91: An unrelated reference is quoted here, please check it further.
5.Lines 118: "upon HCl addition" is unnecessary because it repeats the expression "HCl was added" in the preceding text.
6.Lines 157 and 160: “references (Table 1)” should be “references in Table 1”.
7.Lines 228: The first letter of ‘the’ should be capitalised.
8.As described in the manuscript, the GLC project reduced the sand proportion in the newly reclaimed farmlands while increasing the silt proportion. Please further explain the reasons for this phenomenon.

Reviewer 2 ·

Basic reporting

Although the topic is of interest to the scientific community, this paper should be improved before being considered for publication in any academic journal. Authors should reconsider the main objective of the paper according to its content. They should try to synthesize and emphasize the main findings of the review contents and avoid long sentences. Additionally, the conclusion is poorly written and fails to summarize the findings and effectively highlight their significance.

Experimental design

The data visualization program employed is Origin 2025, which is appropriate for fundamental plotting. Contemporary research could be enhanced by sophisticated visualization methods such as heatmaps, three-dimensional plots, or interactive graphs utilizing software like R.

Validity of the findings

- Abstract: The authors should revise the abstract; it is too general. In addition, it could be further developed. The article contains a lot of interesting data. An informative and representative conclusion should be added to the abstract.
- Introduction; The research mentions the correlation between soil organic carbon (SOC) and clay content and the effect of bulk density (BD) on water retention and nutrient storage, but it does not address how soil properties vary across spatial (local, regional, global) and temporal (seasonal, long-term) scales. Studying how these qualities change across time and land-use scenarios is possible.
- There is also limited exploration of how soil properties and their interactions vary across different ecosystems or climates.

Additional comments

- In the main text, many numeric data are given with too many significant figures; 2 significant figures suffice, and 3 suffice in case the first significant figure is "1".
- You must provide all the figures in high resolution. Make all the labels and legends more legible.
- Conclusion: The findings could be further developed, as there is a lot of interesting data in the article.
- Limitations: It is necessary to include the possibility that your study is affected by this study because it is not a community study; please review this bias and include this information in the limitations section. Please include in the limitations that your study can't be extrapolated to other realities. You may recommend that this study should at least be done in the community to contrast the results.

·

Basic reporting

Article title: Dokuchaev's paradigm of genetic soil science, with which Yenni agrees, indicates that soil factors influence processes, and processes determine properties. The authors examine the Gully Land Consolidation Project, which can be seen as an anthropogenic factor, but it is somewhat mechanistic to interpret the influence of factors directly on soil properties without analysing the role of soil-forming processes.

Line 44: ‘Soil properties are critical factors...’ is a clearly erroneous statement. See the argumentation above.

Line 47: ‘Soil properties such as ... soil water content (SWC) ... are closely related to soil carbon storage potential’ - these two properties have a completely different scale of variability over time to be compared with each other.

Line 48-49 - these findings are somewhat naive, as similar counterexamples can be found. The content of organic matter in soil is determined by a complex of factors (soil-forming rock, climate, relief, living organisms and time). It is therefore inappropriate to state individual correlations without context.

Line 51: Similarly, this phrase is meaningless without context.

Line 53: ‘SWC influences soil respiration’ - this is not technically true: soil respiration depends on the activity of biota and the volume and composition of soil air. Water is an antagonist of soil air and can only affect soil respiration through this. The nature of the impact also depends very much on the aggregate structure of the soil, so to say that ‘SWC influences soil respiration’ is to greatly impoverish the real picture of events in the soil.

Line 55: The above is not true for all soils, but for those that the authors think about but do not name.

Line 55: ‘Thus, soil properties are crucial in regulating soil quality and increasing soil carbon sinks.’ - the phrase is both trivial and tautological: soil has only soil properties, and quality is a certain composition of soil properties that people perceive as useful for themselves.

Line 55: ‘Changes in land use and management practices can alter soil properties, soil structure, ...’ - soil structure is a subset of soil properties, so they cannot be listed separated by commas.

Experimental design

The authors mention ‘Gully Land Consolidation’, but it is not clear from the text what exactly it is. This project was launched in 2011, but the authors consider options with different cultivation times, even up to 70 years. The design of the experiment is also not entirely clear.

What is the name of the soils according to WRB?
In the Methods section and in Figure 1, the authors indicate the experimental plots and name them according to the time of cultivation. In Figure 2 and in the discussion of the results, the authors give the geographical names of the objects. It is impossible to compare these methods of designating research objects.

Validity of the findings

The authors interpret the geographical differences between the study sites as a consequence of the influence of cultivation time. However, the organisation of the research cannot resolve this problem: sampling points of the same cultivation time are located nearby and thus spatially atocorrelated, which makes it impossible to apply traditional statistics. The authors obviously do not have enough replicates to apply geostatistics. In addition, the particle size distribution is a rather conservative indicator, which makes it difficult to interpret the differences found by the authors as the result of the influence of factors in such a short time.
The article contains many inaccuracies and minor errors. In principle, they can be corrected, and it would have been desirable for the authors to do so before submitting the article to the journal. But the article also contains fundamental methodological flaws that cannot be corrected. The organisation of the field experiment does not allow, in principle, to solve the problems that the authors set themselves. In general terms, the main drawback of the manuscript is the processing of the results of the replications. It is also worth recommending that the authors pay attention to theoretical issues of soil science before trying to solve undoubtedly important practical problems.

---

## Round 0.2 · accepted · Accept

Dear Dr. Yu, I am pleased to inform you that your article has been accepted for publication. I hope that you will continue to research this important topic and submit articles to our journal again.

Reviewer 1 ·

Basic reporting

The authors revised all the comments

Experimental design

Good

Validity of the findings

Well

Additional comments

good

Reviewer 2 ·

Basic reporting

This revised version of the manuscript is suitable for publication in PeerJ.

Experimental design

-

Validity of the findings

-

Additional comments

-